# Milk-Derived miR-22-3p Promotes Proliferation of Human Intestinal Epithelial Cells (HIECs) by Regulating Gene Expression

**DOI:** 10.3390/nu14224901

**Published:** 2022-11-19

**Authors:** Rulan Jiang, Bo Lönnerdal

**Affiliations:** Department of Nutrition, University of California, Davis, CA 95616, USA

**Keywords:** milk derived-miR-22-3p, microarray, proliferation, CAAT/enhancer-binding protein δ (C/EBPδ)

## Abstract

MicroRNA (miRNA) is small non-coding RNA involved in gene silencing and post-transcriptional regulation of gene expression. Milk exosomes are microvesicles containing microRNAs (miRNAs). miR-22-3p (miR-22) is plentiful in human milk exosomes and may contribute to intestinal development since milk exosomes and microRNAs are resistant to gastrointestinal digestion in infants. After miR-22 mimics were transfected to human intestinal crypt-like epithelial cells (HIECs) using Lipofectamine for 24 h, RNA was isolated for microarray assay. Microarray results show that miR-22 markedly regulates gene expression, and the roles of miR-22 include promotion of proliferation, regulation of immune functions, and inhibition of apoptosis. Based on the microarray results and miR-22 predicted target genes, CCAAT/enhancer-binding protein δ (C/EBPδ) may be an important direct target of miR-22. C/EBPδ is a transcription factor that regulates numerous biological processes including cell proliferation. In miR-22 transfected HIECs, expression of the C/EBPδ gene was significantly inhibited. Silencing of the C/EBPδ gene by siRNA resulted in increased proliferation of HIECs. A luciferase assay showed that miR-22 specifically binds to the 3′-untranslated region of C/EBPδ mRNA. In summary, milk-derived miR-22 promotes intestinal proliferation by modifying gene expression, and C/EBPδ may be an important target for miR-22 involved in this effect.

## 1. Introduction

Human milk contains various bioactive components, including microvesicles (exosomes) that play beneficial roles in infant development [1,2]. Milk exosomes are enriched with microRNAs (miRNAs). These are short noncoding RNAs consisting of 18 to 25 nucleotides that function in gene silencing and post-transcriptional regulation of gene expression by binding to the 3′-untranslated region (3′-UTR) and other regions of target mRNAs [1,3,4]. miRNAs regulate more than 60% of protein coding genes [5]. The phospholipid membrane of exosomes protects microRNAs from ribonuclease digestion, freeze–thaw cycles and acidity [6]. Since milk exosomes and microRNAs are partly resistant to in vitro gastrointestinal digestion and are internalized by normal human intestinal crypt-like epithelial cells (HIECs) [7], they may contribute to intestinal development in infancy.

miRNA hsa-miR-22-3p (miR-22) is one of the most abundant miRNAs in human milk exosomes [7,8,9]. miR-22 is known to be involved in a broad range of biological activities, such as cell proliferation and differentiation [10,11], apoptosis [12,13,14], and immune functions [15,16]. miR-22 is transcribed from the short arm of chromosome 17 and is highly conserved across many vertebrate species, including chimp, mouse, rat, dog, and horse [17]. Currently, functions of miR-22 derived from milk exosomes still need investigation.

CCAAT/enhancer-binding protein δ (C/EBPδ) is a member of the highly conserved C/EBP family of basic region leucine zipper transcription factors, and it influences a variety of biological processes such as growth arrest, cell death, cell proliferation and, cell differentiation, as well as inflammation [18,19,20,21]. C/EBPδ is a potential target gene of miR-22 since its 3′-UTR contains a predicted miR-22 binding site (http://www.targetscan.org, accessed on 1 September 2021).

In the present study, effects of miR-22 on HIECs, a normal human intestinal epithelial cell line [22], were evaluated using microarray and whole genome transcriptome analysis. Based on the microarray results and miR-22 target gene prediction, functions of miR-22 in HIECs and effects on a direct target gene were subsequently investigated.

## 2. Materials and Methods

### 2.1. In vitro Digestion

miR-22 mimics (30 pmole/mL, Sigma-Aldrich, St. Louis, MO, USA) dissolved in infant formula (1 mL, Enfamil, Mead Johnson Nutrition, Evansville, IN, USA) was digested as described previously [23] by sequential treatment with pepsin at pH 4.0 (Sigma-Aldrich, St. Louis, MO, USA) and pancreatin at pH 7.0 (Sigma-Aldrich), each for 30 min at 37 °C. The amount of miR-22 surviving simulated gastrointestinal digestion was evaluated by qRT-PCR using a miScript SYBR Green PCR kit (Qiagen, Valencia, CA, USA) according to the manufacturer’s instructions. Snord96a (Sigma-Aldrich) was used as an internal control.

### 2.2. Cell Culture

Non-transformed human intestinal crypt-like epithelial cells (HIECs), a gift from Dr. Jean-François Beaulieu (Université de Sherbrooke, Sherbrooke, QC, Canada) [22], were cultured in Opti-MEM (Life Technologies Inc., Carlsbad, CA, USA) supplemented with fetal bovine serum (FBS, 5%, MP Biomedicals, Santa Ana, CA, USA), GlutaMax (1%, Gibco, Thermo Fisher Scientific, Grand Island, NY, USA), and epidermal growth factor (EGF, 5 ng/mL, Gibco) and were maintained in a humidified incubator at 37 °C under an atmosphere of 5% CO_2_. Medium was renewed three times per week, and cells between passages 18–24 were harvested at 90% confluence.

### 2.3. miR-22 Transfection and Verification

HIECs in 6-well plates (~80% confluence) were transfected with miR-22 mimics or a negative control (non-targeting miRNA mimics) (30 pmole/well, Sigma-Aldrich, St Louis, MO, USA) using Lipofectamine RNAiMAX (Invitrogen, Carlsbad, CA, USA) following the manufacturer’s instruction. After 24 h of incubation, total RNA was isolated using Trizol reagent (Invitrogen) and intracellular miR-22 was quantitated by qRT-PCR using a miScript SYBR Green PCR kit (Qiagen, Valencia, CA, USA) following the manufacturer’s instructions. Snord96a (Sigma-Aldrich) was used as an internal control.

### 2.4. Microarray and qRT-PCR Verification

HIECs in 6-well plates (~80% confluence) were transfected with miR-22 mimics (Sigma-Aldrich, 30 pmol/ well, 6-well plates) using Lipofectamine RNAiMAX (Invitrogen) following the manufacturer’s instructions. 24 h after transfection, total RNA was extracted from HIECs using Trizol Reagent (Invitrogen). RNA was digested with DNase I (New England Biolabs, Ipswich, MA, USA) and then purified by an RNease kit (Qiagen) according to the manufacturer’s instructions. Total RNA was amplified and labeled with biotinylated nucleotides using a TotalPrep RNA Amplification Kit (Ambion, Austin, TX, USA). Bead chips (Illumina, San Diego, CA, USA) were scanned with the Illumina iScan using standard conditions.

Purified RNA (1 µg) isolated from HIECs was reverse-transcribed to cDNA using a high-capacity cDNA reverse transcription kit (Applied Biosystems, Foster City, CA, USA) according to the manufacturer’s instructions. Quantitative real-time polymerase chain reaction (qRT-PCR) was performed using the cDNA reaction mixture (2 µL) and SYBR Green (Bio-Rad, Hercules, CA, USA) with the iCycler real-time PCR system (Bio-Rad). Gene-specific primers (Table 1) to C/EBPδ, TP53INP, IFIT3, IFIT1, CXCL10, p21, and GAPDH were designed using the primer tool from NCBI (https://www.ncbi.nlm.nih.gov/tools/primer-blast/, accessed on 10 September 2021) and ordered from Operon Technologies (Alameda, CA, USA). The cycling parameters were 95 °C 15 min and 40 cycles including 95 °C for 15 s, 60 °C for 30 s and 72 °C for 30 s. Linearity of the dissociation curve was analyzed, and the mean cycle time of the linear part of the curve was designated as cycle threshold (Ct). Each sample was analyzed in triplicate and normalized to GAPDH using the following equation: fold-change = 2^(Ct Gene-Ct GAPDH)^. Values are shown as the mean fold-change ± standard deviation, relative to the control (set to 1).

### 2.5. Microarray Data Analysis

Raw signal values for probes were exported from Genome Studio (Illumina) and were input to Bioconductor package limma for analysis [24]. Raw signal values were normalized using the norm function built into limma. Pair-wise comparison was carried out to analyze the difference between the control and miR-22 transfected groups, and *p* < 0.05 was considered significant.

### 2.6. Proliferation Assay

After HIECs were transfected with miR-22 mimics or a negative control for 24 h (Sigma, 3 pmol/well, 96-well plates), as described above, effects of miR-22 on cell proliferation of HIECs were evaluated using a bromodeoxyuridine (BrdU) cell proliferation kit (Roche Diagnostics, Indianapolis, IN, USA) following the manufacturer’s instructions.

### 2.7. Immunoblotting

After HIECs were transfected with miR-22 mimics for 24 h, cells were lysed in RIPA buffer (Cell Signaling Technology, Beverly, MA, USA) supplemented with Halt™ Proteinase Inhibitor Cocktail (Thermo Scientific, Rockford, IL, USA), sonicated briefly, and centrifuged at 5000× *g* for 10 min at 4 °C. Proteins were separated by SDS–PAGE and transferred to nitrocellulose membranes. The membrane was incubated with primary antibodies (anti-C/EBPδ antibody, Santa Cruz Biotechnology, Santa Cruz, CA, USA; anti-p21, anti-phospho-CDK2, anti-CDK2, anti-phospho-Rb, anti-Rb, and anti-β actin antibodies, Cell Signaling Technology, Danvers, MA, USA) and then with secondary antibodies (anti-mouse antibody or anti-rabbit antibody, Cell Signaling Technology). Bands were developed with ProSignal Femto ECL Reagent (Genesee Scientific, San Diego, CA, USA) and visualized with the ChemiDoc TM MP Imaging System (Bio-Rad Laboratories, Mississauga, ON, Canada) using the optimal auto-exposure setting. All raw images were analyzed using both ImageLab 6.1 (Bio-Rad) and ImageJ (NIH).

### 2.8. Construction of C/EBPδ 3′-UTR Reporter Vector and Luciferase Assay

A fragment of C/EBPδ 3′-UTR containing the predicted miR-22 recognition sequence was amplified using human genomic DNA (Novagen, EMD Chemicals, San Diego, CA, USA) as template and then subcloned into the SpeI and HindIII sites of the pMIR-REPORT miRNA expression reporter vector (Ambion, Austin, TX, USA). The primers were 5′-GTCAAAGCTTAACGACCCATACCTCAGACC-3′ (Forward) and 5′-GTCAACTAGTAGGGGCGATTTCAAATGCTG-3′ (Reverse). Site-directed mutagenesis was conducted with a QuikChange site-directed mutagenesis kit (Stratagene, La Jolla, CA, USA) following the manufacturer’s instruction. Insertion and orientation of the fragment were confirmed by sequence analysis. The miRNA expression reporter vector (200 ng) and renilla luciferase control plasmid (20 ng, pRL-TK, Promega. Madison, WI, USA) were co-transfected with miR-22 mimics or a negative control (10 pmol for each) to HIECs in 24-well plates using Lipofectamine RNAiMAX (Invitrogen). The transfected cells were harvested 48 h after transfection, and the luciferase activity was measured with the Dual-Luciferase Reporter System (Promega) following the manufacturer’s instructions. Relative luciferase activity was measured by the ratio of reporter (firefly) to control (renilla) activity.

### 2.9. RNA Interference

HIECs (~50% confluence) were transiently transfected with C/EBPδ small-interfering RNA (siRNA) (Forward: 5′-CCACUAAACUGCGAGAGAAtt-3′, Reverse: 5′-UUCUCUCGCAGUUUAUGGtg-3′, 5.0 µg oligonucleotide/well for 6 well plates, 1.0 µg oligonucleotide/well for 96 well plates, Ambion) for 48 h using Lipofectamine 2000 (10 µL/well for 6 well plates, 2 µL/well for 96 well plates, Invitrogen) according to the manufacturer’s instructions. Cells were either used for immunoblotting (cells cultured in 6 well plates) or proliferation assay (cells cultured in 96 well plates) as described above.

### 2.10. Statistical Analysis

Data represent means ± standard deviations from 3 independent experiments. Comparisons between treatments and the control were conducted using one-way ANOVA (Prism Graph Pad, Berkeley, CA, USA). *p* < 0.05 was considered statistically significant.

## 3. Results

### 3.1. miR-22 Survives in vitro Digestion

To simulate gastrointestinal digestion in infants, miR-22 (in 1 mL infant formula, 30 pmole/mL) was digested sequentially with pepsin at pH 4.0 for 30 min and then pancreatin at pH 7.0 for 30 min. qRT-PCR analysis showed that the amount of miR-22 in digested samples was ~40% of that in the undigested samples (Figure 1). This finding that miR-22 is relatively resistant to in vitro digestion is consistent with the result from our previous study [7].

### 3.2. Whole Genome Transcriptome Analysis Reveals That miR-22 Significantly Regulates mRNA Levels in HIECs

To evaluate effects of miR-22 on gene expression in human intestinal epithelial cells, HIECs were transfected with miR-22 mimics. 24 h later, total RNA was isolated for verification using qRT-PCR as described in Materials and Methods and then for microarray assays. mRNA levels of the whole genome were markedly modified in miR-22 transfected HIECs (Appendix A). A total of 608 genes were significantly regulated: 418 genes were down-regulated and 190 genes were up-regulated. The microarray result was subsequently verified by qRT-PCR of five randomly chosen regulated genes: C/EBPδ, TP53INP, IFIT3, IFIT1, and CXCL10. The qRT-PCR results and microarray results were consistent (Figure 2). Functional categorization of miR-22 effects was then conducted in gene ontology (GO) terms (Table 2 and Table 3). The effects of miR-22 are shown in Table 2 and Table 3; the major effects of miR-22 on HIECs are promotion of cell cycle progression, cell growth and proliferation, protection against viral infection, regulation of immune functions, and inhibition of apoptosis.

### 3.3. miR-22 Promotes Intestinal Proliferation

Since growth and proliferation of intestinal epithelial cells is essential for intestine development during early life [25], we focused on investigating the effect of miR-22 on cell proliferation of HIECs. To examine whether miR-22 increases intestinal proliferation, HIECs were transfected with miR-22 mimics or a negative control, and then a proliferation assay was performed. As shown in Figure 3, the transfected miR-22 mimics but not the negative control markedly increased proliferation of HIECs, suggesting that undigested miR-22 from milk exosomes may promote intestinal proliferation in early life.

### 3.4. miR-22 May Promote Proliferation of HIECs by Directly Targeting the C/EBPδ Gene

The effects of miR-22 on HIECs consist of direct effects by binding to 3′-untranslated regions (3′-UTR) of mRNA of target genes and indirect effects by regulating gene expression. Prediction of direct target genes of miR-22 in HIECs was conducted using TargetScan (http://www.targetscan.org, accessed on 1 September 2021). We then compared the potential miR-22 target genes and the modified genes identified by microarray. According to the analysis and comparison, the significantly regulated C/EBPδ gene may be one of the important target genes since C/EBPδ is a transcription factor that regulates transcription of genes involved in cell growth arrest [18]. As shown in Figure 4A, there is a potential miR-22 binding site in the 3′-UTR of C/EBPδ mRNA. To determine if C/EBPδ is a direct target of miR-22, luciferase vectors containing the 3′-UTR or mutated 3′-UTR of the C/EBPδ mRNA were constructed and then were co-transfected with miR-22 mimics or a negative control to HIECs. As shown in Figure 4B, miR-22 mimics specifically bound to the 3′-UTR of the C/EBPδ mRNA, while there was no binding to the mutated form. Next, we designed experiments to determine if miR-22 inhibits transcription and expression of the C/EBPδ gene. After HIECs were transfected with miR-22 mimics or a negative control, both the mRNA (Figure 5A) and protein (Figure 5B) levels of C/EBPδ were significantly decreased by miR-22 mimics but not the negative control.

We also explored whether endogenous C/EBPδ is involved in cell proliferation. C/EBPδ siRNA was transfected to HIECs to silence the endogenous C/EBPδ gene. After inhibition of C/EBPδ expression was verified by immunoblotting (Figure 6A), HIECs transfected with C/EBPδ siRNA were then tested for effects of C/EBPδ on cell proliferation. As shown in Figure 6B, suppression of endogenous C/EBPδ expression led to increased cell proliferation. Moreover, effects of suppression of expression of the C/EBPδ gene on the RNA and protein levels of cyclin dependent kinase inhibitor 1A (p21) were examined. p21 is a growth inhibitor [26] and a target gene of the C/EBP family [27] as well as a predicted C/EBPδ target gene [28]. p21 protein binds to the CDK/cyclin complex and inhibits its activity, thereby inhibiting phosphorylation of retinoblastoma protein (Rb) and then arresting the cell cycle [29]. After HIECs were transfected with miR-22, both the RNA and protein levels of p21 were decreased (Figure 7A,B), suggesting that the inhibited expression of C/EBPδ caused by miR-22 leads to down-regulation of p21, a growth inhibitor, and thus activation of CDK/Cyclin and Rb phosphorylation as shown in Figure 7B. In summary, miR-22 may directly bind to the 3′-UTR of C/EBPδ mRNA, inhibiting expression of C/EBPδ and its downstream pathways and thereby promoting proliferation of HIECs.

## 4. Discussion

miR-22 is partly resistant to in vitro gastrointestinal digestion and may thus contribute to intestinal development in early life. Milk miRNAs appear in free form and are also packed in vehicles, such as exosomes, milk cells and milk fat globules [30]. miRNAs encapsulated in exosomes are known to be resistant to gastrointestinal digestion *in vivo* [31,32] and in vitro [7,33]. Pancreatin used for in vitro digestion is a mixture of several digestive enzymes including amylase, trypsin, lipase, ribonuclease and protease. Lingual lipase plays a role in lipid digestion in newborn infants due to the immature gastrointestinal tract [34]. Since milk exosome associated miR-22 was not investigated in the current study, in vitro digestion of miR-22 with lingual lipase was not conducted. The undigested miRNAs are then internalized by intestinal epithelial cells and regulate gene expression [7]. Milk miRNAs have been documented to play roles in intestinal maturation, immune response, and suppression of inflammation [35,36,37]. Previous studies have shown abundant levels of miR-22 in both term and preterm human milk [7,8,9]. Since miRNAs packed in exosomes are conserved among mammals [38], miR-22 from both human milk and bovine milk [39] may exert beneficial effects in infants. However, cow’s milk is not recommended for infants before the age of 12 months due to some health risks including excessive protein and total energy intakes [40], and infant formula based on cow milk undergoes heat-treatment [41] which is likely to render miRNAs inactive.

Milk miR-22 remarkably regulates gene expression in human intestinal epithelial cells. In the current study, whole genome microarrays were used to investigate functions of miR-22 in human intestinal epithelial cells (HIECs), which are normal crypt-like intestinal epithelial cells isolated from gestational 14–20 weeks fetal intestines [22]. Similar to effects of miR-22 on gene expression, it was recently reported that transfection of milk abundant miRNAs including miR-3126 and miR-3184 to undifferentiated Caco-2 cells significantly influences whole genome gene expression [9]. Expression of around 600 genes was significantly modified by miR-22 in HIECs. Based on functional analysis of the modified genes, major effects of miR-22 on human intestinal epithelial cells include promotion of proliferation, protection from viral infection, and regulation of immune functions. miR-22 serves as both a tumor suppressor and an oncogene [42] in various cell types [17]. It has been documented that miR-22 suppresses cell proliferation of a variety of cancers, such as colorectal cancer [43], lung cancer [44], glioblastoma [45], and retinoblastoma [11]. miR-22 is also an oncogene [46,47], and it silences a tumor suppressor, PTEN (phosphatase and tensin homolog deleted from chromosome 10) [47]. In the current study, we show that milk exosome-derived miR-22 promotes growth of normal intestinal epithelial cells by regulating gene transcription and inhibiting expression of the C/EBPδ gene. Our result is consistent with literature showing that milk-derived exosomes promote growth of intestinal epithelial cells [48,49,50].

Milk miR-22 directly binds to the 3′-UTR of the C/EBPδ mRNA to mediate intestinal proliferation. By comparing the microarray results and predicted miR-22 target genes, we discovered that the C/EBPδ gene might be an important target gene of miR-22. The C/EBP family consists of six members: C/EBPα, -β, -δ, -ε, -γ, and -ζ; they share a highly conserved, basic-leucine zipper (bZIP) domain at the C-terminal, which is responsible for dimerization, DNA binding, and initiation of gene transcription. Binding of C/EBP to the consensus sequence 5-T (T/G) NNGNAA (T/G)-3′ may initiate gene transcription [51]. C/EBPδ can also form heterodimers with the related protein CEBPα. C/EBPδ is a transcription factor known to participate in the regulation of many genes associated with the cell cycle and immune functions [18]. Numerous studies have reported that C/EBPδ induces growth arrest and functions as a cancer suppressor [21,52,53]. It has been reported that C/EBPδ causes G0/G1 proliferative arrest in chronic myelogenous leukemia cells (KCL22 and K562) by down-regulating c-Myc and cyclin E and up-regulating p27 (Kip1) [52], and C/EBPδ is induced in growth arrested mammary epithelial cells by activating genes related to cell cycle regulation, including septin 7, regulator of chromosome condensation I, and DIRAS family, GTP-binding RAS-like 3 [53]. In addition, C/EBPδ is abundantly expressed in healthy pancreatic ductal cells but not in pancreatic ductal adenocarcinoma. Reactivation of C/EBP δ in pancreatic cancer cell in vitro resulted in decrease of cell proliferation [21]. p21 is a potential target gene of C/EBPδ [28], and it has been extensively studied as a key player in suppressing cell cycle progression and cellular proliferation due to its contribution to the G1/S checkpoint [26,54]. In the present study, miR-22 was shown to directly bind to the 3′-UTR of the C/EBPδ mRNA to thereby enhance proliferation of HIECs by suppressing not only expression of the C/EBPδ gene but target genes of C/EBPδ in intestinal epithelial cells, such as p21. Numerous miR-22 target genes related to cell cycle regulation and immune functions have been identified, such as TGFBR1 [55], BCL9L [56], bone morphogenic protein 7 (BMP7) [57], and histone deacetylase HDAC4 [58].

In summary, milk derived miR-22 is relatively resistant to in vitro gastrointestinal digestion in infancy and may exert various beneficial effects including promotion of cell proliferation, protection against viruses, and regulation of immune functions. C/EBPδ may be an important target gene for miR-22 in intestinal epithelial cells. Thus, miR-22 promotes intestinal proliferation by regulating gene expression and by directly inhibiting expression of the C/EBPδ gene.

## Figures and Tables

**Figure 1 nutrients-14-04901-f001:**
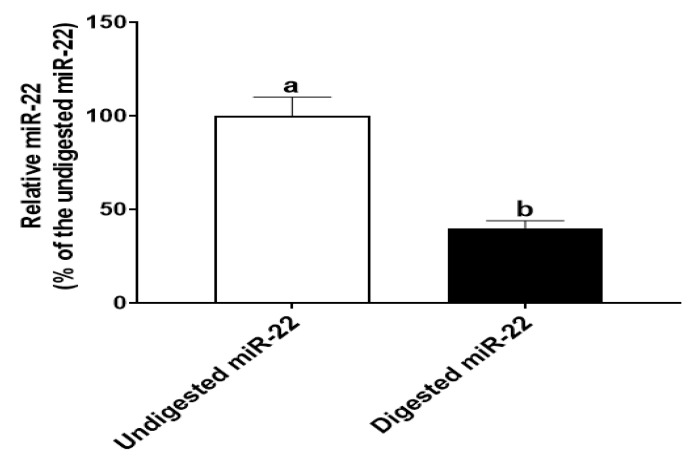
In vitro digestion by HIECs. miR-22 (in 1 mL infant formula, 30 pmole/mL) was digested with pepsin (pH 4.0) and pancreatin (pH 7.0) for 30 min, respectively. After that, miR-22 was quantitated by qRT-PCR. *n* = 5. Values are shown as mean fold-changes ± SD, relative to the control (set to 1). Bars without a common letter are significantly different (*p* < 0.05).

**Figure 2 nutrients-14-04901-f002:**
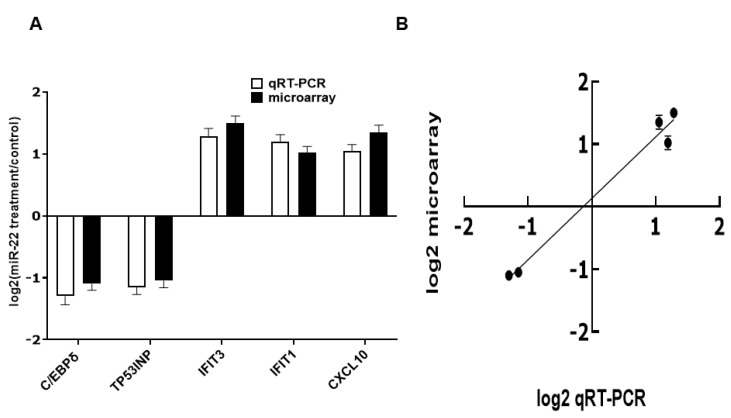
Real-time PCR verification of microarray results. (**A**) Results of qRT-PCR and microarray. (**B**) Correlation between results from microarrays and qRT-PCR. *n* = 5. Values are shown as mean fold-changes ± SD, relative to the control (set to 1). R^2^ = 0.9755.

**Figure 3 nutrients-14-04901-f003:**
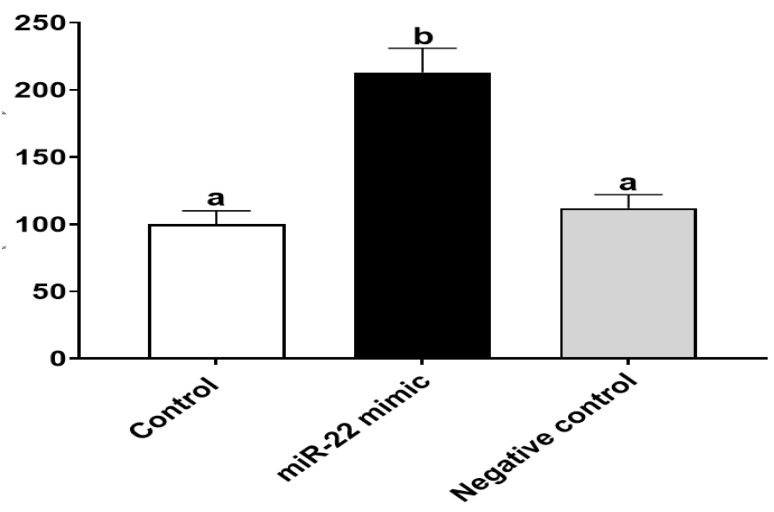
miR-22 promotes proliferation of HIECs. After HIECs were transfected miR-22 for 24 h, BrdU proliferation assays were conducted. *n* = 5. Values are shown as mean ± SD, relative to the control (set to 1). Bars without a common letter are significantly different (*p* < 0.05).

**Figure 4 nutrients-14-04901-f004:**
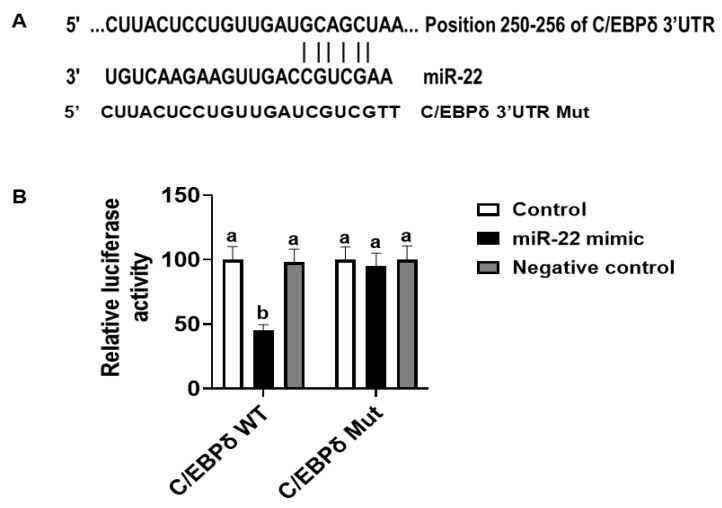
Luciferase assay. (**A**) There is a putative miR-22 binding site at the 3′-UTR of the C/EBPδ gene (TargetScan, accessed on 1 September 2021). (**B**) Constructed luciferase vectors and miR-22 mimics or a negative control were co-transfected to HIECs and then luciferase assays were conducted. *n* = 5. Values are shown as mean ± SD, relative to the control (set to 1). Bars without a common letter are significantly different (*p* < 0.05).

**Figure 5 nutrients-14-04901-f005:**
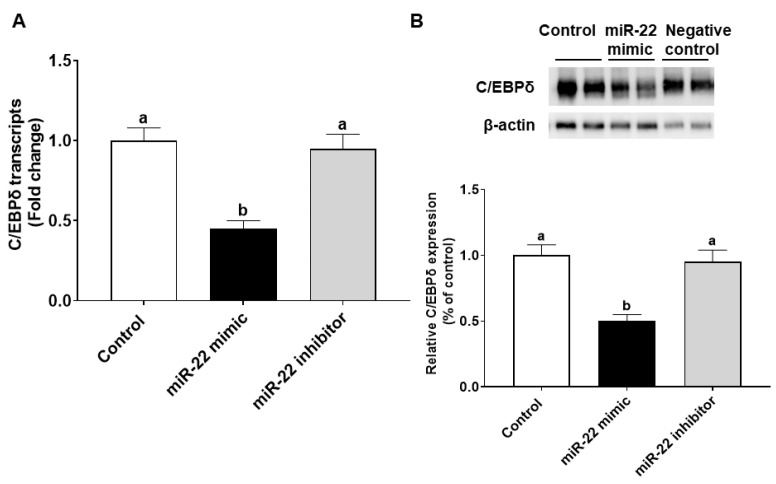
miR-22 inhibits C/EBPδ expression in HIECs. mRNA (**A**) and protein (**B**) levels of C/EBPδ were decreased by miR-22 as revealed by qRT-PCR and immunoblotting results, respectively. Values are shown as mean ± SD, relative to the control (set to 1). Bars without a common letter are significantly different (*p* < 0.05).

**Figure 6 nutrients-14-04901-f006:**
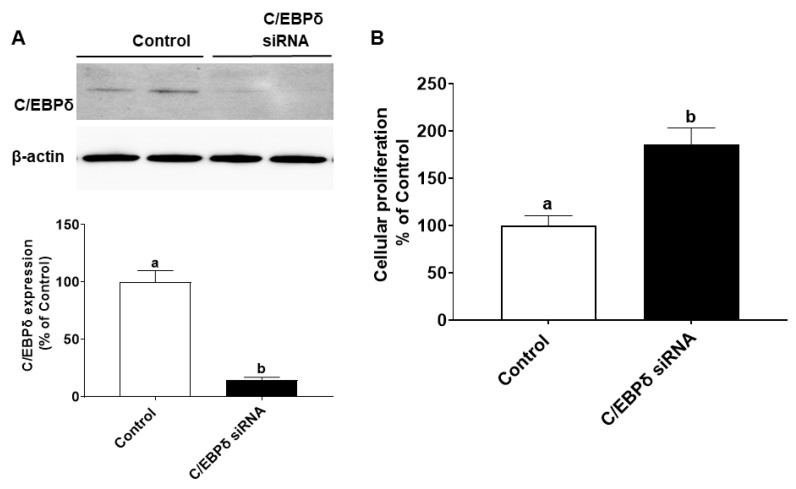
Effects of C/EBPδ inhibition on proliferation of HIECs. HIECs were transfected with C/EBPδ siRNA, and proliferation was then evaluated with a BrdU cellular proliferation kit (Roche). (**A**) C/EBPδ siRNA significantly inhibited the protein level of C/EBPδ. (**B**) After C/EBPδ was down-regulated, proliferation of HIEC cells was markedly increased. *n* = 5. Values are shown as mean ± SD, relative to the control (set to 1). Bars without a common letter are significantly different (*p* < 0.05).

**Figure 7 nutrients-14-04901-f007:**
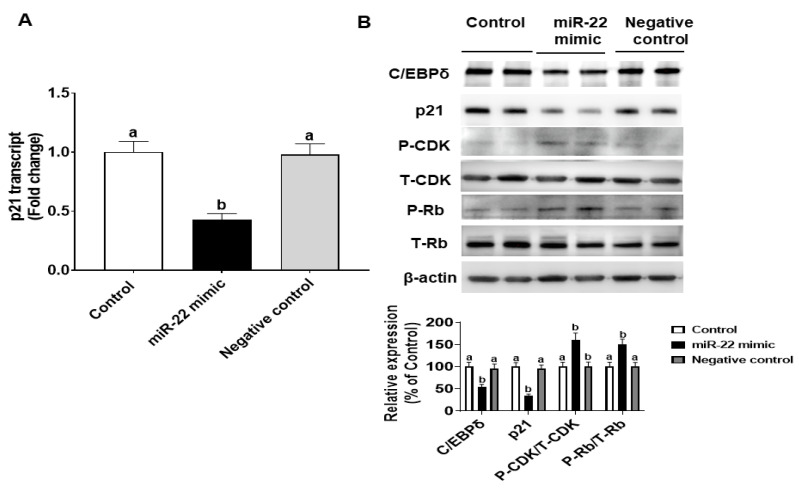
Effects of miR-22 on the protein level of a downstream target of C/EBPδ, p21. After HIEC cells were transfected with miR-22, the RNA level of p21 was evaluated by qRT-PCR (**A**,**B**) Protein levels of p21, phospho-CDK2 (P-CDK2), total-CDK2 (T-CDK2), phosphor-Rb (P-Rb), and total-Rb (T-Rb) were evaluated by immunoblotting, respectively. *n*= 5. Values are shown as mean ± SD, relative to the control (set to 1). Bars without a common letter are significantly different (*p* < 0.05).

**Table 1 nutrients-14-04901-t001:** Sequences for qRT-PCR primers.

	Primer Sequence,	5′-3′	
Gene	Forward	Reverse	Accession Number
C/EBPδ	GCCATGTACGACGACGAGAG	GAGTCGATGTAGGCGCTGAA	NM_005195.4
TP53INP	CACCCGTGGGACTGATGAAT	GAGCTTCCACTCTGGGACTAC	NM_033285.2
IFIT3	TTTGAAGCAGGCCATTGAGC	CAGGCCCAAGAGAACCTTGA	NM_001549.2
IFIT1	AACCTCGTCATTGTCAGGCA	TCCGCTGCCCACTTAGAGAA	NM_022168.2
CXCL10	CCTTAAAACCAGAGGGGAGCA	TGTGGTCCATCCTTGGAAGC	NM_001565.1
p21	ACTTTGGAGTCCCCTCACCT	CCCTAGGCTGTGCTCACTTC	NM_000389.5
GAPDH	GCTGAGTACGTCGTGGAGTC	AAATGAGCCCCAGCCTTCTC	NM_001289746.2

**Table 2 nutrients-14-04901-t002:** Functional enrichment analysis in GO terms (upregulated).

Term	Annotated	*p* Value
G1/S transition of mitotic cell cycle	200	0.00019
Regulation of mitotic cell cycle phase transition	218	0.0008
Positive regulation of cell proliferation	505	0.0019
G2/M transition of mitotic cell cycle	151	0.00264
Positive regulation of cell cycle process	109	0.00012
Transmembrane receptor protein tyrosine kinase signaling pathway	668	0.00224
Cell division	276	0.0072
Positive regulation of gene expression	630	0.0087
Defense response to virus	128	7.1 × 10^-5^
Negative regulation of viral genome replication	25	7.2 × 10^-5^
Type I interferon signaling pathway	44	0.00026
Positive regulation of type I interferon production	37	0.00429
NIK/NF-kappa B signaling	59	0.01305
Positive regulation of interleukin-6 production	15	0.0199

**Table 3 nutrients-14-04901-t003:** Functional enrichment analysis in GO terms (downregulated).

Term	Annotated	*p* Value
Positive regulation of cell death	228	0.00021
Regulation of epithelial to mesenchymal transition	22	0.00161
Positive regulation of apoptotic process	218	0.00214
Regulation of release of cytochrome c from mitochondria	16	0.00216
Positive regulation of protein insertion into mitochondrial membrane involved in apoptotic signaling pathway	13	0.00237
Negative regulation of cytoskeleton organization	55	0.00487
Regulation of reactive oxygen species metabolic process	53	0.00806
Negative regulation of protein phosphorylation	120	0.022
Negative regulation of cell communication	410	0.0288

## Data Availability

All data are available upon request to the corresponding author.

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
