# Peer review of "Milk-Derived miR-22-3p Promotes Proliferation of Human Intestinal Epithelial Cells (HIECs) by Regulating Gene Expression"

_nutrients, 2022, doi:10.3390/nu14224901_

Round 1
Reviewer 1 Report
The Jiang and Lonnerdal article explores a component of genomic regulation in exosomes when is included in formula, and how this compound may affect the biology of gastrointestinal cells. They present very interesting and curious data. In general, miR-22 up-regulates cell development and proliferation genes and down-regulates apoptosis genes and ROS. It is really a very well written, understandable and relevant article. However, I would like to comment on a few contributions:
1. The miR-22 is combined with formula to perform the experiments, this is interesting from the industrial point of view, since, the formulas do not have this compound. However, it would be interesting to test it in breast milk, have these experiments been considered in human samples?
2. The introduction is very well written and follows very well.
3. The material and methods are very reproducible. I wonder if it is necessary to include a salivary or intestinal lipase process in the digestion process, since in neonates, salivary lipase plays an essential role.
4. On the other hand, fig 2 is not very understandable, perhaps it should be moved to a supplementary figure, since in the text it is mentioned how many genes are up- or down-regulated. Fig 3B, I do not think it is relevant, in general, it is consistent that both techniques correlate, isn't it?
Author Response
We greatly appreciate the constructive and valuable comments from the reviewers. We have conducted further experiments and revised the manuscript according to the comments.
The Jiang and Lonnerdal article explores a component of genomic regulation in exosomes when is included in formula, and how this compound may affect the biology of gastrointestinal cells. They present very interesting and curious data. In general, miR-22 up-regulates cell development and proliferation genes and down-regulates apoptosis genes and ROS. It is really a very well written, understandable and relevant article. However, I would like to comment on a few contributions:
1. The miR-22 is combined with formula to perform the experiments, this is interesting from the industrial point of view, since, the formulas do not have this compound. However, it would be interesting to test it in breast milk, have these experiments been considered in human samples?
Human milk already contains free forms of miRNAs and exosome associated miRNAs, and it would be difficult to assess any further effects by addition. Therefore, to simulate miRNAs in human milk, miR-22 mimics were combined with infant formula in the current study.
- The introduction is very well written and follows very well.
- The material and methods are very reproducible. I wonder if it is necessary to include a salivary or intestinal lipase process in the digestion process, since in neonates, salivary lipase plays an essential role.
Pancreatin used for in vitro digestion is a mixture of several digestive enzymes including amylase, trypsin, lipase, ribonuclease and protease. Salivary lipase may play a role in digesting milk exosomes in newborns. Only miR-22 mimics were used in the current study, so effects of salivary lipase were not evaluated. This information has been included in the revised manuscript.
- On the other hand, fig 2 is not very understandable, perhaps it should be moved to a supplementary figure, since in the text it is mentioned how many genes are up- or down-regulated. Fig 3B, I do not think it is relevant, in general, it is consistent that both techniques correlate, isn't it?
Figure 2 has been removed from the revised manuscript, and the results are shown in Supplementary Figure 1. Fig 3B shows that the microarray and qRT-PCR results are consistent, which is a verification of the microarray results and therefore is important.
Reviewer 2 Report
The hypothesis tested by the study is interesting but the data presentation must be improved.
For ex: in the figure legends is never specified if these are the results from independent experiments and how many of them. Image resolution is poor and it is difficult to read the graphs. The backgrounds of the immunoblotting has to be consinstent through all the figures. In fig. 8b the b actin does not seem to correspond to other markers. Please definitively improve figure legends.
What it is the specific population the authors refer to: newborn or infant?
There are several spelling errors through the manuscript. Please, read it thoughtfully
Author Response
We greatly appreciate the constructive and valuable comments from the reviewers. We have conducted further experiments and revised the manuscript according to the comments.
The hypothesis tested by the study is interesting but the data presentation must be improved.
For ex: in the figure legends is never specified if these are the results from independent experiments and how many of them. Image resolution is poor and it is difficult to read the graphs. The backgrounds of the immunoblotting has to be consistent through all the figures. In fig. 8b the b actin does not seem to correspond to other markers. Please definitively improve figure legends.
All the Figure legends have been revised. We stated that “Data represent means ± standard deviations from 3 independent experiments” in the Materials and Methods section. Sample size for each experiment has been added.
What it is the specific population the authors refer to: newborn or infant?
Infants including newborns.
There are several spelling errors through the manuscript. Please, read it thoughtfully
The errors have been corrected.